# Artificial Intelligence Technologies Revolutionizing Wastewater Treatment: Current Trends and Future Prospective

Ahmed E. Alprol [1], Abdallah Tageldein Mansour [2,3,*], Marwa Ezz El-Din Ibrahim [4,5] and Mohamed Ashour [1,*]

1   National Institute of Oceanography and Fisheries (NIOF), Cairo 11516, Egypt
2   Animal and Fish Production Department, College of Agricultural and Food Sciences, King Faisal University, P.O. Box 420, Al Hofuf 31982, Saudi Arabia
3   Fish and Animal Production Department, Faculty of Agriculture (Saba Basha), Alexandria University, Alexandria 21531, Egypt
4   Department of Food and Nutrition, College of Agriculture and Food Sciences, King Faisal University, P.O. Box 420, Al-Ahsa 31982, Saudi Arabia
5   Department of Nutrition and Food Science, College of Home Economics, Helwan University, Cairo 11611, Egypt
*   Correspondence: amansour@kfu.edu.sa (A.T.M.); microalgae_egypt@yahoo.com (M.A.)

**Abstract:** Integration of the Internet of Things (IoT) into the fields of wastewater treatment and water quality prediction has the potential to revolutionize traditional approaches and address urgent challenges, considering the global demand for clean water and sustainable systems. This comprehensive article explores the transformative applications of smart IoT technologies, including artificial intelligence (AI) and machine learning (ML) models, in these areas. A successful example is the implementation of an IoT-based automated water quality monitoring system that utilizes cloud computing and ML methods to effectively address the above-mentioned issues. The IoT has been employed to optimize, simulate, and automate various aspects, such as monitoring and managing natural systems, water-treatment processes, wastewater-treatment applications, and water-related agricultural practices like hydroponics and aquaponics. This review presents a collection of significant water-based applications, which have been combined with the IoT, artificial neural networks, or ML and have undergone critical peer-reviewed assessment. These applications encompass chlorination, adsorption, membrane filtration, monitoring water quality indices, modeling water quality parameters, monitoring river levels, and automating/monitoring effluent wastewater treatment in aquaculture systems. Additionally, this review provides an overview of the IoT and discusses potential future applications, along with examples of how their algorithms have been utilized to evaluate the quality of treated water in diverse aquatic environments.

**Keywords:** Internet of Things; artificial intelligence; machine learning; artificial neural network; soft sensors for water-treatment plants



## 1. Introduction

The most important component of human existence and industrial operations is water, which is currently seriously threatened by dangerous contaminants brought on by both human activity and natural processes. Water accessibility in safe and healthy ways is a big issue across the world. Therefore, it is vital to categorize and keep track of the water quality; however, the fundamental problem is that, with current technology, adequate parametric quality metrics are not accessible [1]. Due to the increase in human population, the activities of aquatic systems (aquaculture, aquaponics, and hydroponics) have increased. As a result, the nutrient load, mainly nitrogen and phosphorus, drained to the water bodies has been increased, causing damage in several water habitats [2,3]. Therefore, the treatment of effluent aquaculture wastewater should be improved and developed in a sustainable way using modern technologies [4,5].

Water- and wastewater-treatment facilities, as well as numerous industrial and biological systems that depend on different resources, must have access to sustainable and clean water. Treatment facilities must deal with complicated regulatory procedures to fulfill the rising standards of quality, in addition to catering to customer wants and enhancing infrastructure for quality of life [6]. Approximately 300–400 million tons of contaminants are reported to be released into global water every year, leading to water pollution, which is a great burden on water quality management [7]. This is only complicated by the fact that nations continue to have severely contaminated rivers, which damage aquatic and terrestrial life in addition to human life. These problems are gradually worsening as nations continue to industrialize and modernize [8]. Researchers from all over the world have looked for ways to improve water-related applications [8–11]. For several years, there has been enough focus on developing and modeling optimal, economical, and intelligent models to help resolve this problem [12,13].

The integration of the IoT into the areas of water treatment and water quality prediction has the potential to revolutionize traditional approaches and address pressing challenges. In these fields, the current review article explores the transformative applications of the smart IoT, including AI models and ML methods.

While AI involves empowering algorithms to perform tasks and make inferences that would typically require human intelligence, ML focuses on intelligent systems that can adapt their behavior based on new information provided during the training phase [14]. The concept of AI revolves around enabling algorithms to perform tasks and make deductions that would typically necessitate human intelligence. On the other hand, ML is centered on intelligent systems that possess the ability to adjust their behavior in response to newly presented information during the training phase [15]. The adoption of AI and ML in academic communities across various fields and industries is primarily driven by their ability to enhance and facilitate understanding. This extends to research areas such as water treatment, including coagulation and chlorination dosing, membrane-filtration modeling, adsorption processes, natural system monitoring like river quality modeling, and agricultural system health. Previous studies have indicated that ML can serve as an effective approach to address the challenges encountered in these domains [16].

The increased utilization of AI and ML in academic communities spanning diverse fields and industries is largely motivated by their capability to augment and streamline the process of comprehension. This encompasses research endeavors related to water treatment, such as coagulation and chlorination dosing, membrane-filtration modeling, adsorption processes, and the monitoring of natural systems, including river quality modeling and agricultural system health. Previous studies have demonstrated that ML can be a valuable strategy for overcoming the challenges encountered within these domains [13]. The growing adoption of AI and ML in academic communities across various fields and industries is driven by their ability to enhance and streamline the process of understanding. This applies to research areas like water treatment, which include different activities, such as coagulation and chlorination dosing, membrane-filtration modeling, adsorption processes, and the monitoring of natural systems like river quality modeling and agricultural system health. Previous studies have provided evidence that ML can serve as a valuable approach to address the challenges faced in these specific domains [17]. The generality, resilience, and relative simplicity of the design of ML, AI, and smart technologies in water applications would enable them to model and resolve difficult and complicated situations to reduce costs and improve operations [18].

Previous research has shown that the implementation of AI models as effective tools in the water-treatment fields has produced excellent results. The majority of review articles on AI applications in water treatment, however, concentrate on specific areas of water-treatment techniques or process designs, such as membrane bioreactors, membrane processes, adsorption processes, and water-treatment plants. Furthermore, there are not many review articles that provide a thorough introduction to the popular AI models used in water treatment, including their benefits, drawbacks, and recommendations. Thus, this

review paper discusses the performance and applications of artificial neural networks (ANN) and ML in several waters of surface water, groundwater, drinking water, wastewater, and oceans, along with the advantages and disadvantages of commonly used ML techniques. This review article provides a cross-section of critically important water-based applications that have been combined with the IoT, ANN, or ML. These applications include chlorination, adsorption, membrane filtration, monitoring of water quality indices, modeling of water quality parameters, monitoring river levels, and automation/monitoring of aqua-systems' effluent water treatment, in addition to giving a short overview of both water infrastructure resiliency improvement and creating soft sensors for water-treatment plants. In addition, this review discusses the overview and the possible future applications of the IoT and the instances where their algorithms have been used to assess the quality of treated water in various aquatic environments.

## 2. Smart Technology

Computer systems that can learn from data without being explicitly programmed are referred to as ML, a subfield of AI that focuses on the development of algorithms and statistical models (Figure 1).

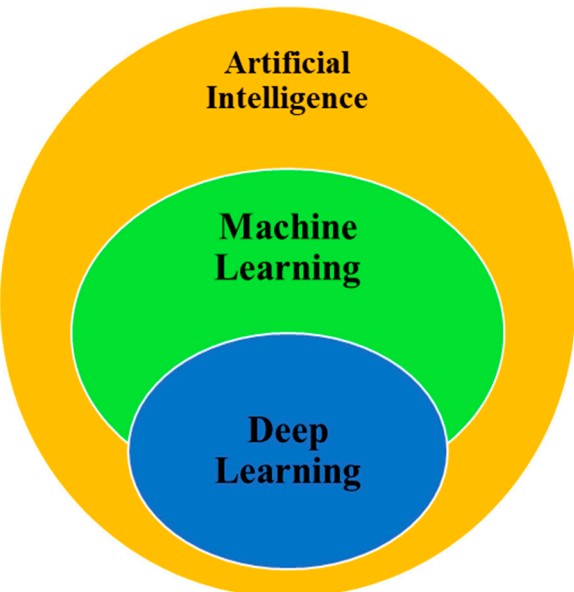

**Figure 1.** Artificial intelligence methods.

Developing predictive models with the ability to make precise predictions or decisions based on data-driven learning is the fundamental aim of ML. Finding patterns or generating predictions from large amounts of data generated by different scenarios is a common use case for ML, a powerful data analysis technique. Before ML is applied in a practical setting, it is necessary to complete data collection, appropriate algorithm selection, model training, and model validation [19]. The main differentiating factor between these two types is the presence of labels in the datasets. The term "Internet of Things" refers to a network of physical objects that can connect to the internet or other communication networks. These objects are often equipped with software, hardware, or other technologies that enable them to facilitate analytical processes, such as environmental sensing. In water applications, internet-enabled systems incorporating sensors for pressure, flow, or water quality/characteristics are commonly employed [20].

During the lifetime of the sensor or that of other technologies, the objective is typically to exchange data with other connected devices or networks, optimize the system, and open up the system or usability [21]. Using labeled training datasets, supervised learning is used to build predictive functions. Input values and anticipated output values are included in

every training instance. To create a predictive model that would forecast the outcome based on the relevant input data, supervised learning algorithms look for correlations between input and output values. The most often used ML models include ANFIS, KNN, DT, SOM, PCA, SVM, RF, and PSO [22].

Often used to identify patterns or forecasts based on massive data generated by multiple situations, ML is a powerful technique for data analysis. Using labeled training datasets, supervised learning is used to build predictive functions. Input values and anticipated output values are included in every training instance. In order to create a predictive model that would forecast the outcome based on the relevant input data, supervised learning algorithms look for correlations between the input and output values [23]. An outline of the ML models used in water applications is given in Section 3. Also, it provides a brief explanation of the employed AI methodologies. The smart sensors, Internet of Things, and systems built using these technologies, which are typically combined with AI/ML models and methodologies, are all covered in Section 3 on smart technologies [24]. All of these methods have been researched for use in hydroponic and aquaponic farming, as well as procedures for treating water and wastewater, including chlorination, adsorption, and membrane filtration. They have also all been studied for managing water quality, including dissolved oxygen and water levels. The subfield of ML that includes deep learning is essentially three-layer neural networks. By "learning" from vast volumes of data, these neural networks aim to mimic the activity of the human brain, albeit far from approaching its capacity. A neural network with only one layer can still generate educated guesses, but it can be optimized and refined for accuracy with the help of more hidden layers [25]. Many AI services and apps rely on deep learning to increase automation by carrying out physical and analytical operations without the need for human participation. Both established products and services (including digital assistants, voice-activated TV remote controls, and credit card fraud detection), as well as cutting-edge innovations (like self-driving automobiles), are powered by deep learning technology.

## 3. AI Models

According to the literature, Figure 2 depicts the important AI models used in WWT. These models can be divided into three categories: ML, ANN, and SA.

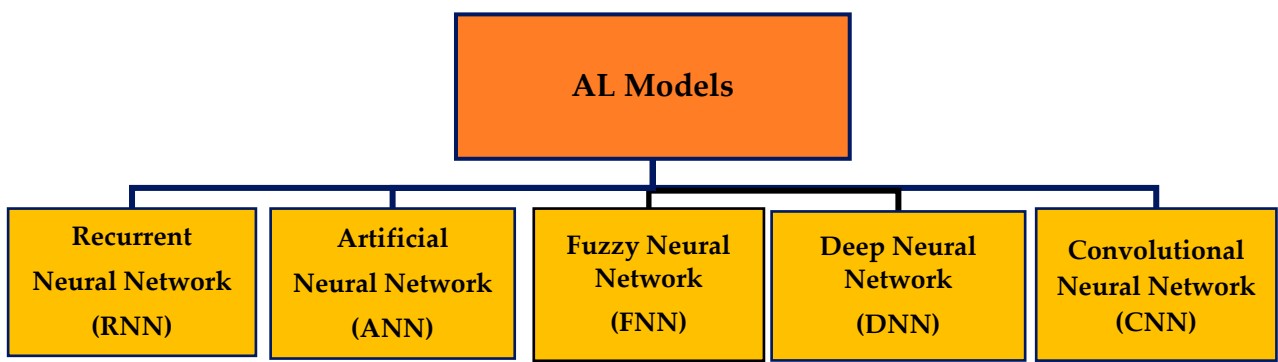

**Figure 2.** Models of AI.

### 3.1. RNN

An RNN, a type of NN, operates by iterating in the direction of sequence evolution and taking sequence data as input. RNN has memory, parameter sharing, and Turing completeness, making it advantageous for learning nonlinear features in time-series problems (Figure 3A). The LSTM is the most commonly used RNN to address the problem of gradient disappearance [26]. RNN has achieved notable success in various applications, such as water, WWT, water quality management, and water-based agriculture.

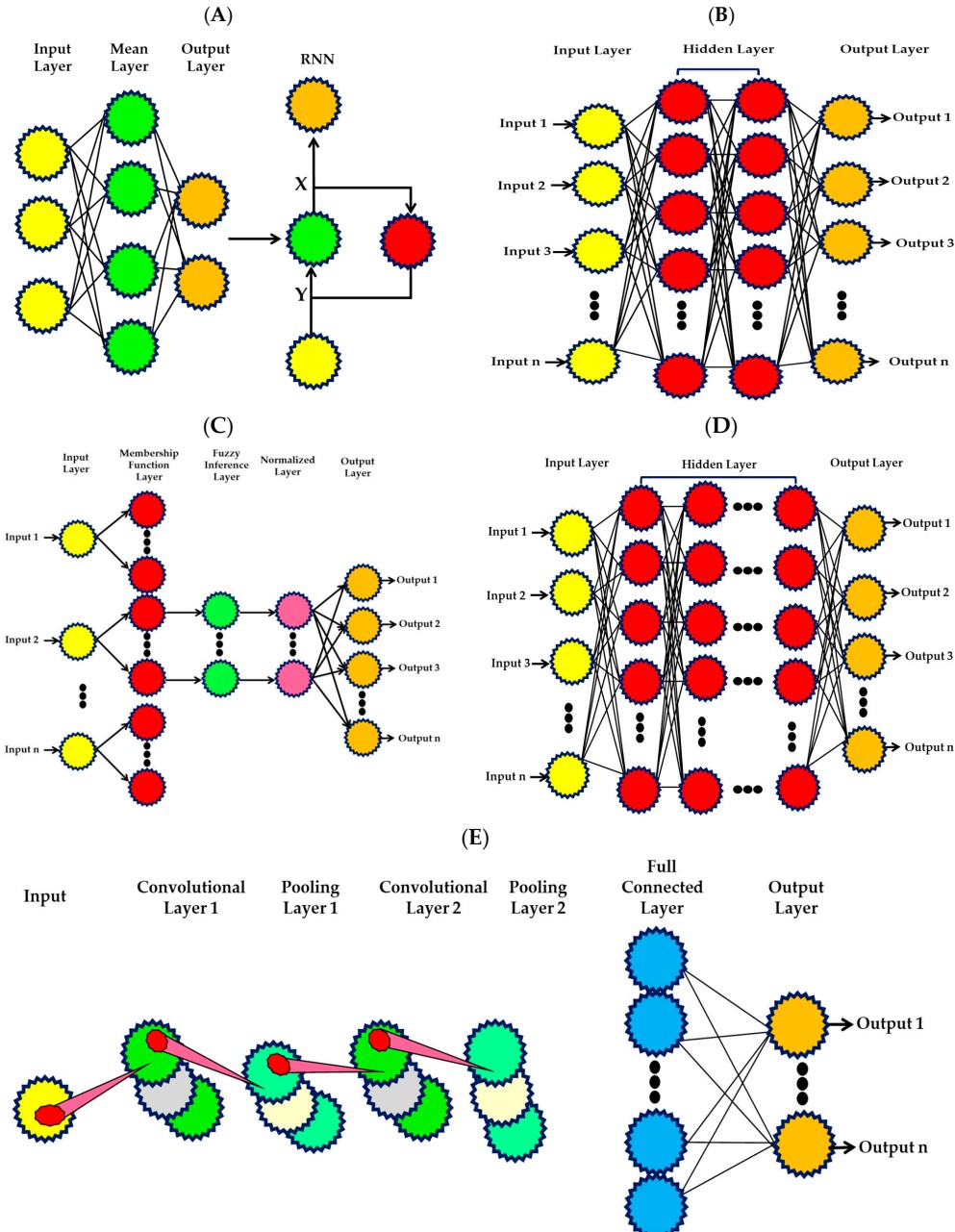

**Figure 3.** Basic planning of ANN models. (**A**) RNN; (**B**) ANN; (**C**) FNN; (**D**) DNN; and (**E**) CNN.

### 3.2. ANN

ANNs are mathematical models that simulate the behavior of biological neural networks (BNNs) to process data. ANNs utilize unit nodes to mimic neurons and perform information processing by adjusting the interconnection weights among multiple nodes in the network. The typical components of an ANN include an input layer, an output layer, and several hidden layers situated between them (Figure 3B). To perform complex nonlinear computations, ANNs employ various activation functions, such as sigmoid, tanh, and ReLU functions, along with multiple variable weights between neurons. Increasing the number of hidden layers in an ANN enables the creation of more intricate nonlinear models and enhances their expressive capabilities [27]. ANNs can be trained to understand complex nonlinear relationships between inputs and outputs by formulating and optimizing loss functions. Recurrent neural networks (RNNs), CNNs, FNNs, and DNNs are the most commonly used ANN models [28].

### 3.3. FNN

FNN, a hybrid NN model, combines ANNs and fuzzy logic to handle uncertain or ambiguous situations. FNN utilizes fuzzy logic reasoning to process input data and then employs an ANN to train and produce the results (Figure 3C). The structure of FNN resembles that of a conventional NN, but it incorporates fuzzy logic to define the fuzzy relationship between inputs and outputs, as well as the weights of connections among neurons (normalization, fuzzy inference, and membership function). The FNN architecture comprises an input, membership function, fuzzy inference, normalization, and output layers. FNN offers significant advantages over standard NNs in dealing with challenging problems and finds extensive use in pattern recognition and control [29].

### 3.4. DNN

DNN is a type of ANN that consists of multiple hidden layers positioned between the input and output layers. The deep architecture of DNN allows it to learn hierarchical data representations by combining lower-level characteristics in successive layers to learn higher-level features (Figure 3D). Similar to other neural networks, DNNs can have a larger number of hidden layers and neurons and are commonly used to capture complex patterns in data or learn highly nonlinear mappings from inputs to outputs. However, training DNNs can be challenging and computationally expensive due to their intricate network architecture, which requires a large amount of data for training [30].

### 3.5. CNN

CNNs are a type of deep learning technique that utilizes convolutional computations and a deep structure feedforward neural network. CNNs are widely used in computer vision, natural language processing, and other fields. CNNs employ convolutional layers to extract intricate features from input images, pooling layers to reduce the feature of dimensionality, and fully connected layers for classification or regression tasks [31]. A typical CNN architecture includes input, output, convolutional, pooling, and fully connected layers (Figure 3E).

## 4. Methods of ML and AI

Table 1 provides a comprehensive overview of various AI and ML models and methods used in water treatment and monitoring. It highlights their general usage and specific applications in water treatment and modeling, as well as their advantages and limitations, aiding in the selection of appropriate models and approaches for monitoring applications and water treatment. For more detailed and foundational descriptions of these methodologies and models, additional textbook sources are available.

**Table 1.** An overview of the ML models and AI techniques used in water treatment and monitoring.

| ML and AI Techniques | WWT and Monitoring Applications | Applications | Advantages | Disadvantages | Refs. |
|---|---|---|---|---|---|
| ANN-General | Modeling of dissolved oxygen concentrations, control, and classification of hydroponic systems; dosage and set point of chlorine; calibration of adsorption process parameters; and modeling of membrane-process parameters. | Supervised ML, regression, and classification. | High-dimensional datasets may be handled. Modeling and prediction outcomes are provided in a timely manner. Forward propagation allows for low-cost, quick processing. For more specific benefits of ANN models, see below. | High computational requirements, particularly during the backward propagation stage. Some models and architectures may be complex to understand independently, and there are specific drawbacks associated with ANN models. | [32] |

**Table 1.** *Cont.*

| ML and AI Techniques | WWT and Monitoring Applications | Applications | Advantages | Disadvantages | Refs. |
|---|---|---|---|---|---|
| SVM/SVR | Three things are modeled: the dissolved oxygen content of rivers, the growth rates of aquaponic plants, and the stages in which plants grow—models of chemical and biological oxygen demand (BOD and COD) for membrane-process parameters. | Supervised ML in classification, regression, and pattern analysis | High-dimensional datasets can be handled, meaning there are more inputs than outputs. Small dataset modifications can be handled. Works with linear and nonlinear data. | Comparatively long training times. Modeling needs great computational power production. SVM/SVR is generally not appropriate for larger datasets. | [33] |
| RF | Modeling the percentage removal in the adsorption process. Modeling dissolved oxygen in simple and hybrid ways. | Regression, classification, and machine learning. | Relatively stable with minimal influence of noise and outliers. Capable of managing continuous and categorical inputs, even with missing values/data. | Decision tree "density" determines accuracy and robustness. Model complexity, model training time, and needed computing power all significantly rise with density. | [34] |
| FIS | Hydroponics system, environmental, and chlorine dosage set point control. | AI. Decision-making and system control. | The use of fuzzy logic, as opposed to binary logic, better represents how people make decisions. A clearly defined framework allows for an easy interpretation of results and choices. | Terminology can be misunderstood without knowing fuzzy logic. Application is reliant on operator-specified parameters and is susceptible to human error. | [35] |
| CNN | DBP formation modeling | Regression, classification, segmentation, and supervised machine learning | Results are frequently viewed as being quite accurate. Results from the parallel model are acquired rapidly. Excels at problem-solving using visual inputs. | Model and architecture are comprehensive and intricate in and of themselves. Considerable computing power is necessary. | [36] |
| RNN/LSTM | Simulation of membrane-process parameters and simulation of dissolved oxygen concentrations. | Supervised ML, regression, and Classification. | Suitable for modeling and time-series datasets. Suitable for modeling and sequential datasets. The length of dataset inputs is not constrained. | Training is challenging due to the high computing requirements and the massive and diverse dataset requirements. | [37] |
| ELM | Dissolved oxygen concentration and modeling. | Supervised ML, regression, and classification | Short training times. Appropriate for pattern categorization. | Frequently encounters over-fitting or under-fitting if too many or too few concealed nodes are used. | [38] |

## 5. Machine Learning and Artificial Intelligence Techniques in Numerous Water- and Wastewater-Treatment Applications

Several water- and wastewater-treatment applications have investigated AI and ML approaches. The intelligent design systems of WWT and its reuse can benefit from the use of AI models in conjunction with traditional techniques and IoT architecture. AI models are a valuable and powerful instrument for the prediction, modeling, and optimization of the wastewater-treatment process. They have been extensively utilized in various aspects of WWT, including the elimination of colors, heavy metals, organic materials, solids, microbial contaminants, medication, nutrients, and pesticides from water [39].

The primary fields in which AI models are used are process design and laboratory-scale research. Process parameter optimization and process performance prediction are typically included in process design in real-world applications. Three popular treatment methods that are frequently applied in wastewater and water-treatment facilities are summarized in this section. Most of the input data included in the publications under review were obtained and distributed via standard collection methods by staff members of treatment plants or other regulatory agencies. Integrating intelligent technology with assessed AI

approaches or ML models helps mitigate the issue of data collection. Certain ML models will most likely become more accurate with more data. Finally, this is only a quick summary of the current research interests and does not aim to represent the whole breadth of research into AI and ML applications in the water-treatment industry [21].

While ML models are useful for simulating disinfection by-product (DBP) concentrations and important parameters for adsorption and membrane-filtration operations, AI approaches have demonstrated their efficacy in controlling chlorination. The coefficient of correlation ($R^2$), mean average error (MAE), coefficient of determination ($R^2$), mean square error (MSE), root mean square error (RMSE), and relative error (RE) are frequently used to evaluate the results. Each AI model has benefits, disadvantages, and potential application areas. The most popular AI models for water purification are included in Table 2, along with their uses, benefits, and disadvantages.

**Table 2.** Common AI models for wastewater treatment, their purposes, advantages, and disadvantages.

| AI/ML Models | Objectives | Advantages | Difficulties | Refs. |
|---|---|---|---|---|
| FNN | Prediction, regression, and classification | Suitable for difficult nonlinear problems; simple to implement and comprehend | Complex model architecture and high cost of computation | [40] |
| PSO | Clustering, regression, and classification | Strong universality, high computational efficiency, and simplicity and ease of use | Defective for discrete issues and sensitive to beginning circumstances | [41] |
| CNN | Regression, segmentation, and classification | Suitable for modeling photos and extraction of key characteristics from images | Computationally costly and difficult to learn | [42] |
| RF | Prediction, regression, and classification | Easy to use and simple, and suited for high-dimensional datasets | Costly to compute, requires thick decision trees to ensure correctness and robustness | [42] |
| RNN | Regression, prediction, and classification | Appropriate for time-series modeling | Computationally costly and challenging to train | [43] |
| DT | Regression, classification, and optimization | No requirement for processing beforehand, and it is simple to comprehend, interpret, and classify | Unsuitable for uneven datasets and ineffective training | [44] |
| PCA | Clustering | Reduces dimensionality, is simple and straightforward to use | Loss of some crucial information and sensitivity to noise in the data | [45] |
| DNN | Prediction, regression, and classification | Rapid and accurate forecast- Appropriate for difficult nonlinear problems | No requirement for processing beforehand, and it is simple to comprehend, interpret, and classify | [42] |
| SVM | Regression, prediction, and classification | Able to solve situations with huge dimensions and appropriate for complicated separable datasets | Costly in terms of computation and unsuitable for bigger datasets | [42] |

## 5.1. Disinfection and Chlorination By-Product Management

Disinfection is a crucial process in water- and wastewater-treatment plants that involves eliminating bacteria and viruses by utilizing chlorine-based disinfectants. Although chlorination is effective at disinfection, it also poses health hazards. Table 3 applied machine learning models for controlling the water's chlorine content. Chlorine can react with organic compounds and bromide in water systems, leading to the formation of disinfection by-products (DBP) [46]. These DBPs are considered potential human carcinogens and repro-

ductive disruptors, prompting increased global research interest [47]. ML technology holds promise in predicting and mitigating the generation of DBPs in drinking water, as well as controlling their levels using AI technologies [48]. Numerous researchers have conducted model testing on surface waters treated with chlorine as the primary disinfectant in drinking water plants, while some studies have explored pre-chlorination peroxide/ozonation. Successful estimation of DBP concentrations in treated water distribution networks and consumer residences has been achieved by considering factors such as water temperature, pH, contact time, chlorine concentration, and TOC concentrations as model inputs [49]. ANNs have been extensively tested as ML models for chlorination and DBP prediction, with additional applications utilizing support vector machines, fuzzy inference systems, and genetic algorithms [50]. ANNs frequently beat both GAs and SVMs in comparative tests, while there are some circumstances when SVMs provide a minor edge when $R^2$ is utilized as a point of comparison [51]. The most well-tested ML model to predict chlorination and DBP is the artificial neural network (ANN). Fuzzy inference systems, support vector machines, and evolutionary algorithms are used in other applications. Total trihalomethanes (TTHM) and total haloacetic acids (THAA) are common DBPs that have been modeled and/or predicted. Certain DBP compounds, such as dichloroacetic acid (DCAA), trichloromethane (TCM), bromochloroacetic acid (BCAA), bromodichloromethane (BDCM), trichloroacetic acid (TCAA), and dibromochloromethane (DBCM), have been the subject of specific studies [52].

**Table 3.** Using machine learning and artificial neural networks to control the amount of chlorine in water.

| ML/AI Method Used | Target Substance and Disinfectant | Input Parameters | Output | Ref. |
|---|---|---|---|---|
| BDCM and TCM | Chlorine | UVA254, temperature, pH, and $Cl_2$ concentration | DBP tap concentration | [53] |
| ANN | Free residual chlorine set point (FRC) | Production flow rate of the WTP, set point output of the reservoir, FRC of the treated water tank, compensatory system flow rate, FRC output of the WTP (mg/L), and dosage error | WTP FRC set point, chlorine dosage | [54] |
| ANN and SVM | TTHM and chlorine ($Cl_2$) | Chlorine, pH, temperature, TOC, UV254 | post-monsoon season (PoM) | [55] |
| FIS | Chlorine quantity and chlorine (ClO) | pH, temperature, time, and raw water total organic carbon (TOC) | Chlorine dosage, FRC | [56] |
| RBF-ANN | HAA5, BCAA, and HAA9 | UVA254, dissolved organic carbon, bromine concentration, temperature, pH, $Cl_2$ concentration, $NO_2$-N concentration, temperature, pH, and $NH_4^+$-N concentration | DBP tap concentration | [52] |
| ANN | TTHM and chlorine | Conditions such as temperature, concentration of algae, pH, TOC, amount of chlorophyll-a, post-chlorine, and content of total chlorine | TTHM wastewater content | [57] |
| SVM, RF, and ANN | TCAA and DCAA | The number of aromatic bonds, atomic distribution of electronegativity, and hydrophilicity and electrotopological characteristics related to electrostatic interactions | DBP wastewater content | [58] |
| THAA, TCAA, and DCAA | Chlorine | Fluorescence spectra | DBP wastewater content | [59] |

### 5.2. Adsorption Procedures

Adsorption techniques are widely recognized as physical and chemical treatment alternatives for the removal of various contaminants in water and wastewater treatments. Adsorption involves the transfer of a target molecule (adsorbate) from a fluid to a solid surface (adsorbent) through an exothermic mass transfer process [60]. However, accurately calculating crucial parameters and predicting the performance of the adsorption process can be challenging due to the complex interactions involved [61]. ML models have been used to enhance the adsorption process by providing critical predictions (Table 4). ML can assist operator decisions in adsorption processes and has been used to model and predict parameters in metal-, industrial color-, and organic chemical-contaminated water streams [62]. Common variables in the ML modeling of adsorption processes include the pH, water temperature, adsorbent dose, contact time, and initial adsorbate concentration, which are included in. Other inputs, such as the adsorbent particle size, system flow rate, agitation speed, bed height, and BET surface area, have been utilized in specific models [60].

Studies have investigated the adsorption processes of various organic contaminants with different characteristics, such as the target contaminant molar mass. Most studies focused on the adsorption efficiency and representing the percentage of removed adsorbate. Some models have also attempted to predict the relative importance of the input water-quality parameters, adsorption capacity, and non-dimensional effluent concentrations. Among the ML models, ANN has been frequently used in research with metal, organic, and industrial dye contaminants. SVM, ANFIS, and RF are other models that have shown successful applications. $R^2$ values for the ANN, SVM, and RF ML models often exceed 0.9 and occasionally surpass 0.99, indicating strong performance [63]. In most situations, the SVM models outperformed the ANN models, providing statistically significant $R^2$ and RMSE values. When compared to other success models for adsorption processes, the optimized ANFIS model did well in one instance [64].

**Table 4.** Predictive machine learning models for adsorption processes of various contaminants.

| Adsorbent | Contaminated Target | ML Technique | Input Variables | Ref. |
|---|---|---|---|---|
| Encapsulated nanoscale zero-valent iron | Phosphate | ANN | pH, phosphate concentration, adsorbent dose, stirring rate, and reaction time | [65] |
| Natural walnut-activated carbon | Methylene blue (MB), Cd(II) | ANN | pH, MB concentration, Cd(II) concentration, adsorbent mass, and contact time | [49] |
| Nickle(II) Oxide nanocomposites | Asphaltenes | RF, ANN, and SVM | Initial copper concentration, adsorbent dose, pH, contact time, and the addition of NaNO$_3$ | [39] |
| Typha domingensis (Cattail) biomass | Ni(II), Cd(II) | ANFIS | pH, adsorbent dosage, metal-ions concentration, contact, and biosorbent particle size. | [66] |
| Activated carbon | Various organic pollutants | ANFIS, ANN, and SVM | Initial concentration, molar mass of target contaminant, bed height, specific surface area, flow rate, and contact time. | [64] |

### 5.3. Membrane-Filtration Procedures

Membrane processes are commonly used to remove contaminants that require a high level of removal, especially those that are difficult or costly to eliminate through other means [67]. Popular membrane techniques include ultrafiltration, reverse osmosis, nanofiltration, and microfiltration [68]. Researchers have created reverse osmosis, ultrafiltration, nanofiltration, and microfiltration models. An additional study using a submerged membrane bioreactor is included in this review. Many natural and industrial wastes, including

oil and petroleum, organic materials found in the environment, numerous industrial and pharmaceutical wastes, and plain salt or ocean water, are among the many pollutants and natural chemicals that these models are used to study. As discussed in previous sections on ML, ANNs are the predominant model used in water/wastewater-treatment applications. For simulating the membrane-filtration processes, ANNs such as RNNs (some of which contain LSTM), as well as ANFIS and SVM, have also been utilized [69]. Table 5 showcases the applications that employ ML to model, predict, and enhance the membrane-filtration process.

**Table 5.** The applications that apply ML to model, predict, and improve the membrane-filtration process.

| Parameter | Algorithm | Input Parameters | Refs. |
|---|---|---|---|
| Prediction of DO | BWNN, ARIMA, BANN, and ANN | Dissolved oxygen | [70] |
| Prediction of BOD | RF, DNN, and SVR | Longitude, latitude, time, site actual depth, total coliform, degree of turbulence at sea, temperature, EC, salinity, chlorophyll-a, transparency, density, $PO_4$–P, $NH_3$–N, TP, NOx–N, pH, DO, and TSS | [70] |
| TN and TP prediction | ANN and SVM | Flow travel time, river flow, TN, DO, temperature, and TP | [71] |
| Na, Mg, EC, Cl, $HCO_3^-$, $SO_4$, TDS, and Ca prediction | ANN and SVM | Na, Mg, temperature, EC, $HCO_3$, $SO_4$, pH, Cl, TDS, and Ca | [72] |
| Prediction of algal bloom | ANFIS | TSS, TP, COD, BOD, TOC, DTP, $PO_4$–P, TN, total coliform, $NH_3$–N, $NO_3$–N, chlorophyll-a, temperature, DO, pH, EC, and fecal coliform | [73] |
| Prediction of chlorophyll-a | SVM and ANN | Chlorophyll-a, $PO_4$–P, $NH_3$–N, $NO_3$–N, temperature, solar radiation, and wind speed | [74] |
| Heavy metal contamination assay | PCA | Cu, Cd, Ni, Zn, Mn, Pb, Cr, and Co | [75] |
| Hyperparameter selection optimization | SVR | Chlorophyll-a, EC, BGA-PC, DO, turbidity, fDOM, and pollution sediments | [22] |

Transmembrane pressure, permeate flow, and solute rejection are the three main variables that ML algorithms seek to generate the simulate membrane-filtration processes. Transmembrane pressure, contact/filtration time, flux rate, temperature, pH, and so on are some of the inputs that are provided in part of this published study. Again, it is challenging to fully compare the statistical values obtained by a number of these studies due to the range of models examined for different parameters. Lastly, all the RNN, ANN, and SVM models fared fairly well in terms of the $R^2$ values, regularly achieving values better than 0.9 and, frequently, values higher than 0.99 [76] (Table 5).

### 5.4. Applications in Surface Water

Water quality in metropolitan areas is now declining, mainly as a result of municipal and industrial wastewater produced by human activity. A prominent topic in surface water quality research is the use of ML [77]. Surface water quality can be predicted and analyzed using a variety of methods (Table 6). Much attention has been focused on improving ML model optimization and increasing prediction precision. Gathering data is an essential initial step in building ML models.

Water system management can benefit from integrated and ad hoc water quality monitoring to establish standards. Traditional environmental monitoring techniques are widely used by environmental authorities. However, practical challenges limit the application of standard methods for in situ monitoring [78]. Remote sensing technologies can address

these limitations by revealing migration and distribution patterns of contaminants that are challenging to detect using conventional approaches. They also fulfill the requirements of real-time and extensive water quality monitoring [79]. In a study by Sagan et al. [22], experiment-based ML demonstrated a higher accuracy compared to conventional models for DNN, PLS regression, and SVR. This enables sophisticated optimization based on the combination of real-time monitoring sensor data and satellite data. However, certain aspects of water quality, such as virus levels, may not be easily identified through remote sensing due to the lack of high-resolution hyperspectral data or their optical inactivity. Indirect estimation using other quantitative data may be employed instead.

Based on water image data, Wu et al. [80] created an attentional neural network based on a convolutional neural network (CNN) to distinguish between clean and dirty water. They verified the functionality of their attentional neural network by conducting several comparative tests on a set of pictures of water surfaces. One advantage of CNN is that it may use the reflectance picture as an input straight out of the box without requiring feature engineering or changing any parameters. Some of the collected data may inevitably be inaccurate, corrupted, or incomplete due to technical or human errors; this will produce a sparse matrix and poor performance in model applications. When this happens, data cleaning, another critical step in ML applications, becomes essential [79].

Several strategies can be used to achieve data cleaning, including not using the dataset directly, applying averages or medians, or augmenting the raw data with ML and matrix completion techniques [81]. The features that were utilized to train ML models affect how accurate their predictions are. Redundant variables will make the model more complicated and have a negative impact on the model's inverse power and accuracy. One of the most commonly studied aspects of surface water quality, dissolved oxygen (DO), directly reflects the health of the aquatic environment and its capacity to support aquatic life. The concentration of DO in the Danube River was forecast using the linear polynomial neural network (PNN) model. The BOD, pH, temperature, and phosphorus content were found to be the most significant factors impacting the forecast accuracy among the 17 water quality parameters [82].

The prediction of the DO concentration in St. John's River, USA, is based on five input characteristics: pH, total dissolved solids, chloride, water temperature, and NOx. Of these, pH and NOx have a significant link with DO and can affect prediction accuracy [83]. These results support those of Chen et al. [84], who found that input parameters had an impact on the model's capacity for prediction. Eutrophication is an issue in surface water quality prediction in addition to typical water characteristics.

**Table 6.** Machine learning models applied to surface and drinking water.

| Algorithms | Determination | Input Factors | Ref. |
|---|---|---|---|
| RF, DNN, and SVR | BOD prediction | Site actual depth, latitude, longitude, time, DO, total coliform, temperature, salinity, chlorophyll-a, $NH_3$–N, TP, pH, polychlorinated biphenyls count, NOx–N, $PO_4$–P, and TSS | [13] |
| PNN | DO prediction | Cl–, alkalinity, PO4–P, COD, BOD, pH, temperature, P, $NO_3$–N, and EC | [85] |
| ANN and SVM | TN and TP prediction | DO, TN, river flow, temperature, rainfall, and TP | [13] |
| RF | TRP, TP, $NO_3$–N, and $NH_4$–N prediction | EC, temperature, turbulence, chlorophyll-a, DO, pH, and flow rate | [86] |
| ANFIS | Algal bloom prediction | $NH_3$–N, COD, DTP, PO4–P, TOC, TN, $NO_3$–N, chlorophyll-a, temperature, BOD, flow rate, EC, total coliform, DO, pH, and fecal coliform | [35] |
| SVM and ANN | Chlorophyll-a prediction | Solar radiation, $PO_4$–P, chlorophyll-a, $NH_3$–N, $NO_3$–N, temperature, and wind speed | [87] |
| Attention neural network | Water pollution monitoring | Water images | [13] |

### 5.5. Applications in Wastewater

ML is frequently utilized in wastewater treatment for operations and management of wastewater-treatment plants (WWTPs), technological optimization, and the monitoring and prediction of water quality (Figure 4).

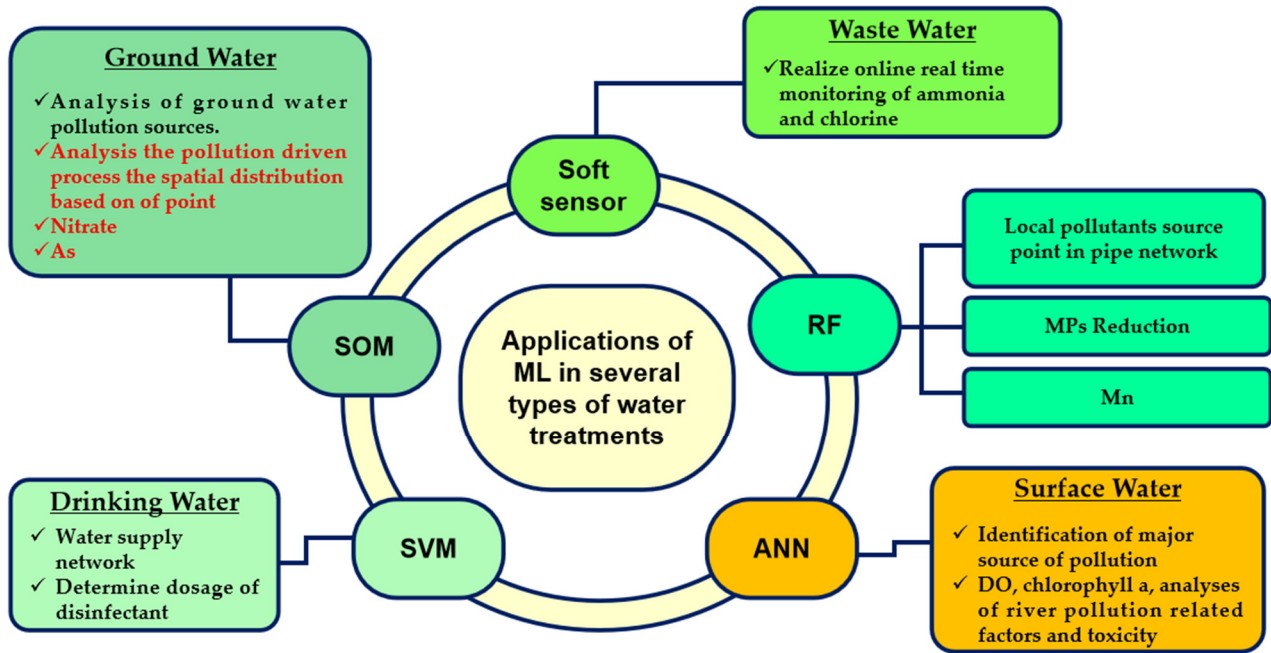

**Figure 4.** Applications of various ML processes in water treatment. (SOM), self-organizing map; (RF), random forest; (SVM), support vector machine; (ANN), artificial neural network; MP, micropollutant.

Pollutants in domestic and industrial wastewater necessitate an assessment of the water's quality prior to treatment [88]. Multiresolution analysis and PCA were combined to create a technique that Rosen et al. [89] found to be more sensitive than PCA for monitoring sewage indicators at various scales. Real-time internet monitoring is crucial for the collection, processing, and analysis of large amounts of data [90]. For example, a soft sensor based on the black-box paradigm was proposed for real-time monitoring of *E. coli* concentrations, revealing significant increases after heavy rainfall, possibly due to urban runoff that resuspends sewer debris [91]. Combining soft sensors with ANNs can help overcome the challenges associated with the expensive and complex operation and maintenance of wastewater-treatment plants, as well as the need for real-time online monitoring of ammonia and chlorine levels [92]. Another example involves the construction of a water quality monitoring system that incorporates a turbidimeter and UV spectrometer for measuring COD and TSP. The system utilized a boosting-iterative predictor weighting-partial least squares (boosting-IPW-PLS) approach with multiple sensors. By assigning weighted parameters and suppressing variables irrelevant to water quality, the boosting-IPW-PLS technique developed a wastewater quality prediction model. Test results showed a significant correlation coefficient between the predicted and actual values, indicating the system's effectiveness in monitoring water quality [93].

A real-world use of ML is in the optimization of wastewater-treatment systems by analyzing historical data. In order to conserve land by condensing the capacity of the anoxic tank, Fang et al. [94] used an SVM and an adaptive evolutionary algorithm to mimic anaerobic, anoxic, and oxic conditions. Additionally, nanofiltration, reverse osmosis, ozonation, and adsorption are tertiary wastewater-treatment methods that have been optimized using machine learning. ML techniques can be widely used in the future for the enhanced treatment of wastewater, including micropollutants (MP) and new contaminants, based on the findings of the aforementioned situations [95]. The work conducted by Bhagat

et al. [13] focused on predicting the removal of copper (Cu) in an adsorption process using attapulgite clay as the primary adsorbent. They developed and compared three models to determine the optimal prediction approach: a RF based on grid optimization, SVM, and ANN. This study was chosen for its detailed description of input selection and model architecture.

To evaluate the impact of additional input variables on the effectiveness of adsorption, the copper ion concentration was maintained at 50 mg $L^{-1}$. The range of values for the adsorbent dosage was 2 to 15 g $L^{-1}$, the pH ranged from 2.0 to 12.0, the $NaNO_3$ concentration varied from 0 to 0.5 mol $L^{-1}$, and the contact period ranged from 5 to 120 min. The RF model, using open-source software in the R programming language known for statistical computation, employed the bootstrapping method. This method splits the data into multiple subsets with random replacement, allowing each decision tree in the RF to have its own random subset for training. In this study, the RF model trained decision trees using 76 samples [13,96].

Each model was tested with different sets of inputs. The input sets included: (1) initial copper concentration; (2) initial copper concentration and adsorbent dosage; (3) initial copper concentration, adsorbent dosage, and contact time; (4) initial copper concentration, adsorbent dosage, contact time, pH, and $NaNO_3$ concentration; and (5) initial copper concentration, adsorbent dosage, contact time, pH, and $NaNO_3$ concentration. Multiple tests were conducted using various input combinations for each model. The input sets for testing included: (1) initial copper concentration; (2) initial copper concentration and adsorbent dosage; (3) adsorbent dosage, initial copper concentration, and contact time; (4) adsorbent dosage, initial copper concentration, pH, contact time, and $NaNO_3$ concentration; and (5) initial copper concentration, adsorbent dosage, contact time, pH, and $NaNO_3$ concentration. The study found that utilizing all five inputs improved the model's performance. Additionally, the SVM model achieved a maximum correlation coefficient of 0.93, while the RF and ANN models exhibited higher accuracy, with correlation coefficients greater than 0.99 [96].

*5.6. Application in Drinking Water*

The application of ML in drinking water management, including source management, distribution, treatment procedures, and decision-making, has proven to be highly beneficial. ML-based approaches provide valuable support in forecasting and assessing source water quality, enabling early detection and the management of pollution (Table 6). Bouamar and Ladjal [97] investigated the effectiveness of multisensor-based ANN and SVM algorithms for the dynamic monitoring of water quality. Both models demonstrated respectable performance in identifying different types of water, with SVM showing greater stability compared to ANN. Wu et al. [98] proposed an adaptive frequency analysis method using drinking-water quality datasets to facilitate an early risk assessment, decision-making, and warning systems for drinking water quality in Norwegian cities. While research on drinking water quality has focused mainly on physical and chemical properties, the microbiological aspects, particularly concerning *E. coli*, have received limited attention [99]. However, the SVM method's simplicity and robustness have made it a popular choice for developing plans related to flocculation and disinfection. Scientists are currently engaged in fault monitoring, disaster prediction, and ensuring the proper operation of urban water-supply system facilities to safeguard the drinking water supply [100]. Despite meeting the necessary criteria at treatment facilities, water can become recontaminated during transit through complex water supply networks. Monitoring biological stability markers and implementing disinfection measures can help address this issue [101]

In terms of control strategies, Wang et al. [102] proposed a predictive control scheme for chemical dosing based on an SVM model that outperformed traditional proportional-integral-derivative feedback control. Cluster analysis has been employed to assess water quality variations in different water networks. Tian et al. [103] utilized cluster analysis to identify the contributions of mixed water sources, such as aluminum migration and sea-

sonal variations, to aluminum residues in urban drinking-water supply systems. Rayaroth et al. [104] introduced a random decision tree bagging classifier using the shuffling frog-leaping optimization method to detect water leaks in distribution networks with optimal sensor placement. The service life of pipelines, crucial for water supply management, was found to be influenced by residual chlorine, and an advanced meta-learning model based on a neural network was proposed by Almheiri et al. [105]. SVM algorithms have also been employed to predict water distribution system contamination events.

Water scarcity is a pressing concern for regional development and population growth. Zhang et al. [106] developed a hybrid statistical model combining ANN and genetic algorithms to predict the performance of drinking water-treatment facilities. This model effectively predicts changes in the water output under various parameter fluctuation scenarios, making it a valuable tool for the efficient management of water-treatment plants. Addressing water supply system imbalances can be facilitated through machine learning approaches. Accurate estimation of water demand is crucial for effective water-resource management, especially in areas with limited water availability [97]. ANN and SVM are commonly used in the drinking water sector, particularly for large-scale applications. Their fast computation time during the training phase enables their utilization of dynamic real-time monitoring systems to ensure the safety and quality of drinking water [97].

### 5.7. Applications in Groundwater

Preserving the security of groundwater is crucial for ensuring human health, as it serves as a significant source of potable water. ML has emerged as a valuable tool for various groundwater analysis tasks, including assessing and predicting groundwater quality and identifying causes of contamination. Multivariate statistical analytic approaches, such as PCA and cluster analysis, have gained wide usage in evaluating groundwater quality. In addition, groundwater quality evaluation has also used ML methods, including SVM, DT, RF, and ANN. The majority of related research on groundwater quality has compared how well different ML algorithms perform when evaluating different problems [107].

Lee et al. [108] analyzed the quality of urban groundwater in Seoul, South Korea, using a combination of fuzzy c-means clustering and a self-organizing neural network. Groundwater samples were categorized into three groups based on contamination levels using a self-organizing map algorithm, and the geographical distribution of these groups was studied to investigate pollution patterns. Jeihouni et al. [109] compared five data mining algorithms, including decision trees, RF, the automatic chi-square interaction detector, and iterative dichotomizer 3, to identify the critical factors affecting groundwater in semi-arid regions and evaluate their impact on high-quality groundwater areas in Tabriz City, Iran. ML has also been used to estimate future water quality and evaluate large-scale regional datasets. Agrawal et al. [110] employed SVM and PSO to estimate and forecast the WQI of groundwater, demonstrating the effectiveness of combining these techniques for groundwater prediction, particularly for pollutants like nitrate and arsenic.

Understanding the causes of groundwater contamination is crucial to ensure its security. PCA and clustering techniques, such as K-means, are commonly used for this purpose in contemporary research [107]. Chen et al. [111] utilized multivariate statistical analysis and PCA to identify the key variables that influence changes in groundwater quality. Decision tree algorithms, commonly used to investigate groundwater resources and quality, can learn the relationships between input and output variables based on specific rules. Random forest (RF) is advantageous in terms of its efficiency and rule generalization ability, making it useful for identifying locations with high-quality groundwater suitable for drinking. RF's performance based on continuous datasets (reaching an accuracy of 97.10%) is particularly noteworthy for groundwater resource planning and management. Integrated models, which combine multiple weak learners into a single strong learner, are widely used to forecast groundwater quality indicators and improve prediction performance. Boosting is an effective integration approach, although precautions must be taken to avoid over-fitting while reducing the variance by combining multiple models [111].

*5.8. Soft Sensing in Wastewater Treatment Facilities*

Since wastewater treatment has little control over the system's input, it is necessary to quickly quantify the process status indications of relevance to enable a prompt response to changing circumstances. The method used to calculate the state indicators can be roughly divided into two categories: primary variables, which are challenging to measure, and secondary variables, which are simple to measure [112]. The secondary factors are simple to measure using a variety of dependable and reasonably priced equipment, as opposed to the more challenging variables. Primary factors depend on a variety of other variables due to the nature of the treatment process, which is driven by the interconnected biochemical events that take place within the system [113]. Also, sensor systems are utilized to monitor wastewater-treatment facilities and maintain plant efficiency and public safety. However, costly or unreliable sensors or off-site laboratory examinations may be needed for effluent parameters of relevance [114]. For instance, ammonium is strictly controlled in discharge water and is a key indicator of treatment effectiveness. However, each ammonium sensor costs more than USD 10,000 [115].

The simple-to-measure variables can be viewed through this lens as inadequate stand-ins for the main process variables. This naturally encourages us to investigate and create an algorithm that, by utilizing a variety of widely accessible secondary variables, can estimate a primary parameter rather precisely. Thus, creating a computer application that may efficiently serve as a soft sensor and a low-cost, easily tuned alternative to pricey instruments was accomplished.

Soft sensors are essentially computer models that seek to precisely estimate the process parameters that are either too expensive for the instrument or that cannot be measured directly [116]. Soft sensors are low-cost alternatives to costly wastewater sensors. They are computer models that precisely predict process factors utilizing the readings from a small number of physical sensors. Soft sensing is recognized for its superior generalization, reduced dependence on subject expertise, and ease of modification to variances [117]. These sensors frequently use data-driven methods that require substantial amounts of annotated datasets and incorporate statistical and machine learning algorithms. Autoregressive integrated moving averages (ARIMA), principal component analysis (PCA), logistic regression, the hidden Markov model (HMM), partial least squares (PLS), random forest (RF), support vector machines (SVMs) and artificial neural networks (ANNs) are some of the methods that are currently in use in the literature [118–121]. The majority of ANN models have been shown to perform better and more reliably than other models.

A specific class of machine learning techniques, known as artificial neural network (ANN) models, consists of a hierarchical structure of neurons, which are numerical universal function approximators. Its well-known moniker, "deep model," is similarly inspired by the hierarchical organization. The field of water-treatment study has developed a number of configurations, or architectures, for the goal of inferring various process variables. Nonetheless, the majority of existing literature has disregarded the temporal notion of processes in the development of inference models [122].

Soft sensors have been developed utilizing modeling approaches, including PCA, PLS, SVMs, and ANNs, for a wide range of uses in the wastewater sector [123]. ANNs with feedforward neural network (FFNN) architecture are the most widely used of these modern approaches [124]. These models have been investigated for the purpose of predicting a variety of process variables: suspended solids (SS), biochemical oxygen demand (BOD), and nutrient removal. Recently, in order to forecast the total nitrogen content, [10] we compared FFNN with support vector machines (SVMs). It is interesting that the study has shown that SVM models perform better than FFNN models. Although FFNN and SVM perform rather well, they are unable to capture the temporal idea of processes by nature and assume that all data are independent. Nonetheless, industrial processes are dynamic in nature, with temporal correlations found in the process data [125]. Therefore, the soft sensor framework makes use of a variety of deep neural network architectures to offer cost-effective monitoring for these facilities as well as reliable predictive modeling.

*5.9. Water Infrastructure Resiliency Improvement*

Water systems serve two distinct purposes, which makes them unique among infrastructure systems. First, these are essential to the supply of water services; as a result, the system's ability to withstand natural disasters and the dangers associated with climate change is inextricably related to the system's ability to produce water [126]. Resilience is the capacity of a water or wastewater system to anticipate, withstand, recover from, and adjust to a variety of challenges related to climate change or other events. Also, resilience is known as a protracted procedure that involves striking a balance between threat and resources and produces adaptable, creative ways to anticipate, manage, react to, recover from, and change in reaction to or before events [127].

Water system management should take into account the following six concepts when selecting resilience measures, in addition to the idea of making decisions under extreme uncertainty [128]: (1) gaining knowledge of the system through critical assessment and network analysis; (2) enhancing maintenance to lower vulnerability and boost resilience; (3) involving users in active demand management; (4) collaborating in nature to manage and respond to risks; (5) creating and refining contingency management, utilizing innovation when necessary. Furthermore, when arguing for resilience investments in water systems, it is important to consider the safeguard services that water systems provide, such as transportation, power, and water itself, since they mitigate the risks related to specific natural disasters [129].

5.9.1. Using AI and ML to Improve Water and Resilient Infrastructure

An essential component of AI and ML success is the exponential development of data. Data are being generated in the modern world at a rate never seen before, and this trend will only increase globally. The increasing accessibility of cloud computing is another important factor contributing to the recent surge in AI and ML. Large AI algorithms may be operated with flexibility and economy thanks to cloud-based platforms, which offer affordable access to the processing power needed. Infrastructure spending is becoming more and more necessary, especially in light of climate change. Our infrastructure is at risk from extreme weather events like flooding and sea level rises; many communities will be left exposed to the effects of climate change if no major investment is made. This involves higher maintenance and operation costs for the infrastructure in a changing environment, in addition to the possibility of infrastructure damage during extreme weather events. AI, ML, and other modern innovations will be essential to these initiatives. Artificial intelligence has a lot to offer that will change the way water infrastructure is planned and managed in the future [130]. This goes beyond simply employing standard pattern recognition or identifying trends from past data. The future of water and infrastructure around the world will be affected by AI in the ways listed below:

a.    Water Quality

Using comparable outcomes and sensor data, AI can examine future water quality patterns to spot changes in the quality of the water that may be a sign of pollution or other problems. In order to address problems such as toxic algal blooms or other pollution, the local agency can now respond with prepared action plans.

b.    Predictive Water Supply Maintenance

AI can help identify maintenance requirements and equipment malfunctions, resulting in increased uptime and less downtime. AI and ML systems that identify possible equipment breakdowns in real-time have been adopted by a number of local governments and agencies across the United States, enabling maintenance crews to take care of problems before they become serious.

c.    Future Forecast for Flood Risk

The natural world may be incredibly intricate when it relates to flooding. The traditional method of estimating future flood risk relied on historical data or "past performance".

AI has given sophisticated predictive modeling a new dimension in detecting flood risks in the future. AI and ML can "learn" using full models rather than simply historical data and "predict" the flood dangers for more complicated places with many risk factors. These types of pilot tests are being carried out all throughout the country. This lessens the impact of floods and enables the responsible authority to take preventative action to safeguard its residents and assets.

d.    Water-Resource and Asset Administration

AI can assist in prioritizing and managing infrastructure assets to guarantee their appropriate upkeep and replacement. It can also lessen waste and improve irrigation, both of which contribute to water conservation.

e.    Energy Efficiency and Sustainability

Water distribution and treatment are prime candidates for AI and ML optimization since they consume large quantities of energy. By examining past trends in water usage, artificial intelligence (AI) can be used to forecast future demands in purification as well as distribution networks. Additionally, the distribution system itself can be optimized with AI. Artificial intelligence (AI) can determine which parts of the network are over- or under-utilized by examining flow rates, pressure, and other data. Adjusting the network and ensuring that water is distributed effectively can act as a decision support system to reduce energy consumption and carbon emissions.

5.9.2. What Role May AI Play in Influencing Urban Water Infrastructure in the Future?

Decentralized, green, circular, carbon neutral, and autonomous are the five main attributes of the future, which are based on the interventions that have been put in place over the past three decades to gradually evolve water systems. The development paths leading to these qualities are connected in terms of improving the system's capacity, performance, and efficiency, all of which eventually help create resilient and sustainable water systems rather than be exclusive [131]. For example, using swales to manage stormwater is a step toward a greener road, but as it can lower energy use and greenhouse gas emissions and disconnect stormwater services from centralized sewer systems, it may also be a step toward the decarbonization and decentralization pathways.

I.    Decentralization pathway

Planning and managing decentralized systems, where local facilities are optimized for water supply, stormwater management, water recycling, and wastewater treatment to prevent needless loss of resources (i.e., water, energy, and materials), can be greatly aided by artificial intelligence (AI). The optimal planning of decentralized systems is a more complex problem than that of centralized systems because these systems need to be linked to form a system of systems that functions as a whole in order to provide the necessary wastewater and water services as well as the greatest possible benefits to the environment and ecology.

Under some circumstances, Garrido-Baserba et al. proposed that economically feasible decentralized systems that do away with the requirement for centralized water supply or collection of waste can be created using existing technologies [132]. However, through the identification of dependable and resilient operations and the creation of predictive maintenance techniques, artificial intelligence (AI) technologies are needed to maximize the various benefits of individual decentralized systems. AI and ML have been used more and more in citizen science projects to improve participation and task allocation. They could be used to facilitate surveys that aim to analyze public perceptions of decentralized systems by finding participants, targeting a particular group of people, and analyzing collected data [131].

II.    Circular economy pathway

WWTPs are the backbone of the circular economy strategy because, in addition to being energy intensive, they account for 25% of the energy used in the water sector, essential

for nutrient recovery, water reuse, and generation of renewable energy using a variety of available technologies [133]. Energy and nutrient recovery from water-treatment systems have been optimized by the application of process-based models [134]. A specific real-world difficulty in highly variable streams of water is driving an increasing amount of implementation of real-time monitoring of crucial process parameters and nutritional characteristics. This enables online learning and control in real-time for optimal resource recovery, as well as the integration of machine learning for precise simulation of system dynamics.

III. Decarbonization pathway

The significance of greenhouse gas (GHG) emissions by water-related activities and the water sector's critical role in mitigating climate change have been emphasized by recent research [133]. The two main causes of greenhouse gas emissions in the water industry are energy use and wastewater treatment. In order to achieve net zero emissions in the water sector, new technologies have been implemented to increase the generation of renewable energy, improve energy efficiency, and reduce emissions from the treatment of wastewater.

IV. Automation pathway

Thanks to advancements in sensing, communication, and computer technologies, automation is widely used in sewage and water treatment plants. Nevertheless, its application is primarily restricted to controlling a single process unit or multiple units within a system [135]. Over the past few decades, studies on the integrated control of UWI systems have advanced quickly, although there are not many practical applications [136]. Enhancing automation or creating completely autonomous water systems presents significant challenges in a number of areas, including the intricate relationships between various system components, the unpredictability of the future, the explainability of control strategies and the credibility of AI, and the financial outlays needed for software and monitoring. For better control performance, machine learning can be included in system control techniques. The benefits of AI integration have been reported to include significantly improved water quality, system reliability, and energy efficiency. A few examples of these benefits include the use of forecasted demands to control water-treatment plant outflows [137]. According to Wang et al. [138], the most recent advancements in deep learning technology have significantly increased the forecasting accuracy for pressure, water demand, water depth, flow, and pollution load. This indicates that system control may be enhanced.

### 5.9.3. Time to Harness the Power of AI and ML

Without question, artificial intelligence (AI) will revolutionize the water and infrastructure sectors in a number of ways. From enhancing the management of water resources to addressing concerns related to climate change and flood resilience, AI and ML have the ability to handle many of the most critical issues facing the sector [130]. We can anticipate even more creative ways to help guarantee a sustainable and effective future for our water and infrastructure systems as technology develops.

## 6. Conclusions and Future Prospective

This review has provided an analysis and overview of various ML models, as well as advanced technologies and techniques applied in different water-related applications. AI and ML approaches have been instrumental in improving, modeling, and automating processes in wastewater treatment, water-based agriculture, and monitoring and management of natural systems. The integration of AI/ML technologies is expected to reduce costs, enhance water-based applications, and offer computer-assisted solutions for complex challenges related to water chemistry and physical/biological processes. ML and AI methods have successfully predicted, modeled, automated, and optimized significant applications in water-related industries and operations, including water- and wastewater-treatment facilities, natural systems, and water-based agriculture. It is recommended to increase future studies on both water infrastructure resiliency improvement (based on water quality) by AI/ML technologies and create soft sensors for water-treatment plants.

However, despite the progress made in research studies, certain challenges and limitations must be addressed. Fully utilizing ML algorithms for water quality evaluation faces several issues, such as:

1.  Data availability and quality: ML often requires a substantial amount of high-quality data. Obtaining sufficient data with high precision is challenging in water-treatment and management systems due to financial or technological constraints.
2.  Limited applicability: ML approaches may not be widely applicable due to the highly complex conditions encountered in real wastewater-treatment and management systems. Therefore, current methods may only be suitable for specific systems.
3.  Data management and legal considerations: Challenges related to data management, public and legal perspectives, repeatability, and transparency in research need to be addressed to further advance intelligent applications in the field.

While these challenges and limitations are evident, ongoing research and development demonstrate the substantial implications and potential of ML, AI, and smart technologies in one of the world's most crucial resources, water.

**Author Contributions:** Conceptualization, A.E.A. and M.A.; validation, A.E.A. and M.A.; project administration, M.A. and A.E.A.; visualization, A.E.A., M.A. and A.T.M.; funding acquisition, A.E.A., M.E.E.-D.I. and A.T.M.; writing—original draft preparation, M.A. and A.E.A.; writing—review and editing, M.A., A.E.A., M.E.E.-D.I. and A.T.M. All authors have read and agreed to the published version of the manuscript.

**Funding:** This work was supported by the Deanship of Scientific Research, Vice Presidency for Graduate Studies and Scientific Research, King Faisal University, Saudi Arabia [GRANT5,483].

**Institutional Review Board Statement:** Not applicable.

**Informed Consent Statement:** Not applicable.

**Data Availability Statement:** Not applicable.

**Acknowledgments:** The authors would like to acknowledge the Deanship of Scientific Research, Vice Presidency for Graduate Studies and Scientific Research, King Faisal University, Saudi Arabia [GRANT5,483].

**Conflicts of Interest:** The authors declare no conflicts of interest.

## Abbreviations

| Abbreviations | Definitions |
| --- | --- |
| WWW | Waste water treatment |
| IoT | Internet of Things |
| AI | Artificial intelligence |
| ML | Machine learning |
| ELM | Extreme learning machine |
| AS | Search algorithm |
| NN | Neural network |
| BNN | Biological neural network |
| ANN | Artificial neural network |
| DNN | Deep neural network |
| BANN | Bootstrapped artificial neural network |
| BWNN | Bootstrapped wavelet neural network |
| ARIMA | Auto-regressive integrated moving average |
| CCNN | Cascade correlation neural network |
| LSTM | Long short-term memory |
| DT | Decision tree |

| SVM | Support vector machine |
|---|---|
| PSO | Particle swarm optimization |
| RF | Random forest |
| GB | Gradient boosting |
| KNN | K-nearest neighbor |
| SOM | Self-organizing map |
| ANFIS | Adaptive-network-based fuzzy inference system |
| PCA | Principal component analysis |
| PLS | Partial least squares regression |
| SVR | Support vector regression |
| DBPs | Disinfection by-products |
| DO | Dissolved oxygen |
| BOD | Biological oxygen demand |
| COD | Chemical oxygen demand |
| TOC | Total organic carbon |
| DTP | Dissolved total phosphorus |
| TP | Total phosphorus |
| TSS | Total suspended solids |
| TRP | Total reactive phosphorus |
| TN | Total nitrogen |
| EC | Electrical conductivity |
| TDS | Tsinghua/Temporary DeepSpeed |
| FDOM | Fluorescent dissolved organic matter |
| BGA-PC | Blue-Green Algae Phycocyanin |
| WQI | Water quality index |

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
