# Peer review of "Artificial Intelligence Technologies Revolutionizing Wastewater Treatment: Current Trends and Future Prospective"

_water, doi:10.3390/w16020314_

Round 1
Reviewer 1 Report
Comments and Suggestions for Authors
First I expect you to answer this question: 1. While there are existing review papers addressing this subject it is crucial to highlight how your study contributes to the field and addresses gaps that have not been covered by previous reviews. It is expected to focus on conducting a comprehensive review of other related reviews in the field to identify the existing gaps as of now, I cannot discern the importance of your research.
2. Please remove all unnecessary self-citations such as the following, I found several of them. As it is a review article it is important to cite only the relevant studies.
“Water and wastewater treatment facilities, as well as numerous industrial and biological systems that depend on different resources, must have access to sustainable and clean water.”
3. The manuscript would benefit from editing by a professional English language editor.
4. In lines 381, 382, and 383, you made a general assumption based on the results of one study and even mentioned the R2 of their model. However, it's important to note that R2 values can vary significantly depending on the dataset, hyperparameters, and other factors. Consider revising this section for clarity.
5. In lines 378 and 379, you mentioned that the SVM model outperformed ANN in most situations. To enhance this discussion, consider creating a table that includes all the papers you reviewed, highlighting the ML models used and their comparative performance in those studies. Be sure to acknowledge the subject of studies, databases, ML targets and inputs, and mention any limitations associated with such comparisons.
6. Add articles focusing on creating soft sensors for water treatment plants. Such as the following study (This is an example you can find any relevant study)
7. For better organization and clarity, consider creating a table that lists all the articles you reviewed in the section on chlorination. Categorize them based on the ML models used, the best-performing ML model in each study, the targets, input variables, and the subject of the study. Ensure comprehensive coverage of a wide range of research areas.
8. Please reconsider the title of Section 5, which currently reads 'Applications of AI and ML in several water and wastewater treatments.' Did you mean 'Applications of AI and ML in several water and wastewater treatment subjects?
9. Add studies focused on water infrastructure resiliency improvement (based on water quality) such as the following study. (This is an example you can find any relevant study)
10. As previously suggested, consider creating tables for all other subsections of Section 5 to provide a clear and organized summary of the reviewed papers and their findings.
Comments on the Quality of English Language
The manuscript would benefit from editing by a professional English language editor.
Reviewer 2 Report
Comments and Suggestions for Authors
This article provides a comprehensive overview of the performance and applications of artificial neural networks and machine learning in various water bodies, including surface water, groundwater, drinking water, wastewater, and oceans. It also highlights the advantages and disadvantages of commonly used machine learning techniques. The authors discusses various aspects of applying the Internet of Things, artificial neural networks, and machine learning in water treatment. These applications include methods such as chlorination, adsorption, membrane filtration, monitoring water quality indices, modeling water quality parameters, monitoring river water levels, and automation/monitoring of wastewater treatment in aquatic systems. In addition, the article introduces the concept of the Internet of Things, discusses its potential future applications, and provides examples of how algorithms can be used to evaluate the quality of treated water in different aquatic environments. The review also summarizes the performance and applications of artificial neural networks and machine learning in surface water, groundwater, drinking water, wastewater, and oceans, as well as the advantages and disadvantages of common machine learning techniques. After reading this article, I have the following suggestions:
1. The article can use more precise expressions such as cloud computing or cloud data to express cloud in line 26.
2. Lack of punctuation in these places( 30 84 97 232 347 383 570)
3. There is no consistency in the use of punctuation in Table 1(34~40)
4. The punctuation of key words is uniform in line 41.
5.The abbreviation of WWT is incorrect in line 44.
6.The expression "equal parameter quality metrics"in lines 54,55 and 56 can be more specific like in lines 56 57.
7.The references in Table 3(in line 402) are not aligned.
8.lines 103 to 111 and 112 to 118 are similar in expression and can be simplified.
9.There has extra spaces in line 169.
10.In line 314, there is a spelling error in the classification in the introduction of SVM's objectives.
Reviewer 3 Report
Comments and Suggestions for Authors
Due to the need to increase the efficiency of sewage and water treatment and numerous scientific works on innovative technological solutions in this area, often based on mathematical models, I consider it important to address the topic of IoT integration in installations/systems whose task is to improve water quality. The presented article comprehensively demonstrates the possibilities of using ML in various research areas, such as water treatment, including coagulation and chlorination, membrane filtration, adsorption processes, as well as river quality monitoring and agriculture. The popularization of the solution based on ML algorithms proposed by the authors may in the future, despite the barriers indicated in the summary, significantly contribute to improving the quality of the broadly understood soil and water environment and have a positive impact on the functioning of living organisms. The article constitutes valuable scientific material and may be published in its current form in the journal.
Round 2
Reviewer 1 Report
Comments and Suggestions for Authors
The authors did not respond adequately to the requested comments, particularly the first comment regarding how their review distinguishes itself from several other reviews on the same subject. Furthermore, despite their assertion that they had removed self-citations of irrelevant articles, they failed to do so. Therefore, I cannot accept the article in its current form.
Comments on the Quality of English LanguageNo comment.
Reviewer 2 Report
Comments and Suggestions for Authors
Now,this article is better thanks before. In my opinion, it could be accepted in this journal.
Round 3
Reviewer 1 Report
Comments and Suggestions for Authors
Thank you.
Comments on the Quality of English LanguageNo comment.